# Stakeholder engagement in a hypertension and diabetes prevention research program: Description and lessons learned

Archana Shrestha[1,2,3]*, Dipesh Tamrakar[1], Bhawana Shrestha[1], Biraj Man Karmacharya[1], Abha Shrestha[1], Prajjwal Pyakurel[4], Donna Spiegelman[2,5]

1 Department of Public Health, Kathmandu University School of Medical Sciences, Dhulikhel, Nepal, 2 Center of Methods for Implementation and Prevention Science, Yale School of Public Health, New Haven, Connecticut, United States of America, 3 Institute for Implementation Science and Health, Kathmandu, Nepal, 4 School of Public Health and Community Medicine, BP Koirala Institute of Health Sciences, Dharan, Nepal, 5 Department of Biostatistics, Yale School of Public Health, New Haven, Connecticut, United States of America

* deararchana@gmail.com

**Data Availability Statement:** All relevant data are within the paper and its Supporting information files.

## Abstract

### Background

Stakeholder engagement is important from the management point of view to capture knowledge, increase ownership, reduce conflict, encourage partnership, as well as to develop an ethical perspective that facilitates inclusive decision making and promotes equity. However, there is dearth of literature in the process of stakeholder engagement. The purpose of this paper is to describe the process of increasing stakeholder engagement and highlight the lessons learnt on stakeholder engagement while designing, implementing, and monitoring a study on diabetes and hypertension prevention in workplace settings in Nepal.

### Methodology

We identified the stakeholders based on the 7P framework: Patients and public (clients), providers, payers, policy makers, product makers, principal investigators, and purchasers. The identified stakeholders were engaged in prioritization of the research questions, planning data collection, designing, implementing, and monitoring the intervention. Stakeholders were engaged through focus group discussions, in-depth interviews, participatory workshops, individual consultation, information sessions and representation in study team and implementation committees.

### Results

The views of the stakeholders were synthesized in each step of the research process, from designing to interpreting the results. Stakeholder engagement helped to shape the methods and plan, and process for participant's recruitment and data collection. In addition, it enhanced adherence to intervention, mutual learning, and smooth intervention adoption. The major challenges were the time-consuming nature of the process, language barriers, and the differences in health and food beliefs between researchers and stakeholders.

**Funding:** • DS, AS, DT • Award no: 5DP1ES02545903 • National Institute of Health • https://www.nih.gov/about-nih • The funders had no role in study design, data collection and analysis, decision to publish, or preparation of the manuscript.

**Competing interests:** The authors have declared that no competing interests exist.

## Conclusion

It was possible to engage and benefit from stakeholder's engagement on the design, implementation and monitoring of a workplace-based hypertension and diabetes management research program in Nepal.

## Introduction

New scientific evidences on healthcare are generated in great volume [1]. However, much of this evidence has been difficult to implement in practice in real settings [2]. Stakeholder engagement can help to address this need by improving the relevance of research by increasing its transparency and accelerating the adoption into real world setting [3]. Stakeholder engagement is also important to generate knowledge, increase ownership, reduce conflict, encourage partnership, facilitate an inclusive decision making and promote equity [4].

Stakeholder engagement is promoted by health research funding organizations as well as researchers to achieve the desired goal [5]. Stakeholders can be engaged across stages of research including identifying topics, choosing hypotheses, analyzing data, and disseminating findings [6, 7]. The levels of involvement range from consultation, to collaboration in bi-directional partnerships and to leading research projects [8].

Despite considerable attention to stakeholder engagement, there is limited information on how stakeholders are engaged. Some reviews describe the engagement in published literature and report impacts of engagement [9–12]. However, there is lack of reporting on how an engagement is implemented. It has been recommended that researchers systematically document and report information about engagement and its impact on individual projects [9, 13]. Therefore, there is the need for demonstrating approaches that have been used to stakeholder engagement and opportunities to learn more about engagement in individual research projects. The purpose of this paper is to describe the process of stakeholder engagement and the lessons learned from this process while designing, implementing, and monitoring a study on diabetes and hypertension prevention in workplace settings in Nepal.

## Materials and methods

### Study setting

Nepal Pioneer Worksite Intervention Study [14] started in 2016 at Dhulikhel Hospital-Kathmandu University Hospital in central Nepal with a goal to prevent diabetes and hypertension among the employees of Dhulikhel Hospital. Dhulikhel Hospital is an independently owned, not-for-profit institution which was conceived and supported by the Dhulikhel community. This study was registered on clincaltrials.gov (Identification Member:NCT03447340; Registration date: 27th February 2018). The study was planned and executed in three phases:

### Phase-1: Formative study

We conducted a formative study to gather data that is useful to develop and implement an intervention program to promote healthy eating at the worksite. The major aim of the formative study was appropriateness to culture and geography to develop meaningful interventions at individual and organizational level. The objective was to define and assess attributes of the study participants, understand the context, tailor intervention at local context, and build relationship between researcher, study populations and other stakeholders.

## Phase-2: Develop culturally appropriate intervention and define research attributes

In the second phase, we used the information from the formative study to develop a culturally appropriate environmental level intervention to promote healthy eating in the cafeteria of the workplace. We further used this information to define study attributes such as primary outcomes, study design, implementation and monitoring strategies, and analysis plan.

## Phase-3: Test the effectiveness of the intervention

In the third phase, we tested the effectiveness of the environmental level intervention using a two-step intervention study. In the first step, we assessed the effectiveness of a 6-month cafeteria intervention on cardio-metabolic risk using a pre-post design. In the second step, we conducted a 6-month, open-masked, two-arm randomized trial by allocating half of the participants to an individual behavioral intervention for the prevention of cardio-metabolic risk in order to estimate the additional effect of an individual level intervention. The design of the study has been published elsewhere [14].

## Stakeholder engagement

**Stakeholder definition.** We defined stakeholder as an individual or group affected by or who can influence the environmental intervention to promote healthy eating in the workplace.

**Engagement objectives.** Our objectives of stakeholder engagement were (a) to identify the specific research needs in their worksite for promoting healthy eating, (b) to develop culturally appropriate healthy eating intervention; (c) to increase stakeholders' participation in study process such as determining study objectives and methods, describing study intervention, recruiting participants, implementing and monitoring the healthy eating intervention; (d) interpreting study results; and (e) enhancing sustainability.

**Identification of stakeholders.** We adopted the 7Ps framework [15] that identifies key groups to consider for engagement. The first, **patients and the public**, represented the current and potential consumers of patient-centered health care and population-focused public health. The second were **providers**, including individuals and organizations that provide care to patients and populations. **Purchasers**, the individuals, and entities responsible for underwriting the costs of health care, such as employers, made up the third group. The fourth group consisted of **payers** who were responsible for reimbursement of medical care, such as insurers. The fifth composed of public **policy makers** and policy advocates working in the non-governmental sector. **Product makers**, representing drug and device manufacturers, comprised the sixth group, and **principal investigators**, or other researchers, made up the seventh. Since this was not a typical patient focus study, but rather a community-based prevention trial in an institutional setting, we modified the definition of the 7Ps according to our context (Fig 1).

**Stakeholder engagement methods.** We involved stakeholders in all three phases.

## At formative study phase

**Formation of study team.** After mapping out the stakeholder for the study, individuals representing each stakeholder category working in nutrition related research were purposively approached in-person and the study team was formed. We selected adult aged 18 years or older, employee of the hospital, who provided informed consent as stakeholder in each category. None of the stakeholders whom we approached declined to participate in the study. The stakeholders were represented as co-investigators and advisors.

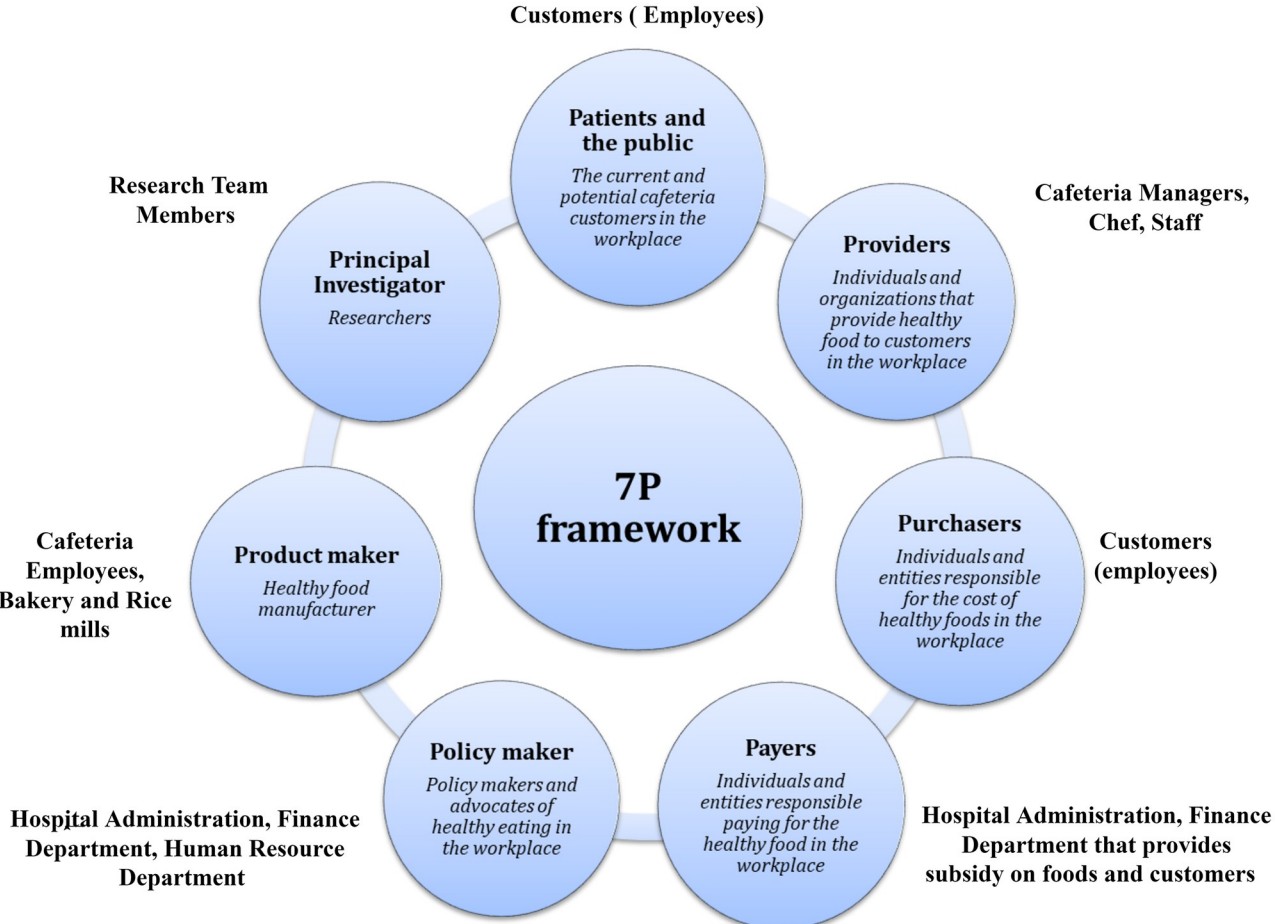

**Fig 1. Stakeholder mapping for the Nepal Pioneer Worksite Intervention Study to prevent cardiometabolic risk.**

**Focus Group Discussions (FGDs).** We conducted four focus group discussions with 33 employees of the hospital to explore the current healthy eating environment, understand drivers and barriers for eating healthy, and to obtain their recommendations on improving environment to promote healthy eating at the hospital cafeteria.

The focus groups were conducted within the workplace setting in a private room to ensure confidentiality and open sharing of opinion. The FGDs were conducted in Nepali and audio taped after obtaining informed consent from the participants. The investigators, AS, or DT, moderated all FGDs in Nepali language and were assisted by a note-taker. In each session, a brief introduction of the study and ethical considerations about maintaining confidentiality of the participants, were explained. The moderator asked open-ended questions and probed for detail information. We used iterative process by discussing each FGD immediately after completion and suggesting necessary detail probing in emerging themes from finding.

**In-depth interviews.** We conducted 9 in-depth interviews with purposively selected hospital employees including a finance manager, a cafeteria manager, an administrative manager and 6 cafeteria operators after receiving informed consent. The investigators DT or AS conducted semi-structured in-depth interviews in Nepali language using a pretested interview guide in a private room of the hospital. We used the iterative process for data collection. After each interview, we discussed each interview and identified the topics to be deeply explored

into themes emerging in earlier interviews. All FGDs and interviews were transcribed verbatim into Nepali by trained research assistants. We used thematic analysis approach [16] to analyze the data.

### At development of research design and intervention phase

**Workshops.** We conducted three iterative workshops with the representation of hospital administration, cafeteria managers, cafeteria staff, nutrition department, employee, and researchers. The aims of the workshops were to consensually decide changes in cafeteria, clarify roles of each stakeholder and develop monitoring strategies. Each workshop was facilitated by DT, a co-investigator, and an employee, using a workshop schedule and discussion guideline. The workshops took 3 hours. In the first hour, the facilitator explained the link between healthy eating and cardiometabolic risks. In the second hour, the findings from the formative study were shared in an interactive session, and in the third hour a guided discussion with the participants was conducted. The themes of the guided discussion were: components of the healthy eating interventions, mode of intervention delivery, mode of implementation and monitoring, and method of evaluation, that enabled us to plan the study better. The stakeholders' in workshops provided inputs in decision making. We presented the initial findings to the stakeholders and collected their feedback.

**Food taste experiments.** A major theme that emerged in focus group discussions, in-depth interviews and workshop included acceptability of introducing whole grains in the cafeteria, especially brown rice. Therefore, the research team decided and conducted a brown rice tasting experiment with 40 employees of the hospital. After tasting brown rice, the participants perception on brown rice changed positively [17].

**Individual consultation.** The results of the formative study with further action plan were discussed individually with the hospital administrator, cafeteria manager, cafeteria chefs, nutritionist of the hospital and few employees of the hospital to get further feedback.

### At testing of intervention phase

**Meeting of study team.** The study team met online every week for 30 to 60 minutes to discuss on a predetermined agenda and all decisions were taken jointly and recorded in written.

**Information sessions.** A 20-minute information session was conducted in all departments of the hospital to provide information regarding the study and collecting feedback. About 22 sessions were conducted with participation of 4 to 20 employees in each session. In the first five minutes, the study design was explained, the next 5 minutes were dedicated for the call for participation and explanations regarding expectations from the participants, and 10 minutes for discussion and feedback.

**Formation of cafeteria intervention committee.** A canteen intervention committee was formed comprising the representations of hospital administrator, human resource department, finance department, hygiene monitoring department, non-communicable disease prevention and management department, nutrition department, canteen managers, employee, and researchers. This committee steered the development, implementation and monitoring of the cafeteria intervention. The committee made the plans for the cafeteria intervention, implementation, and monitoring. It meets once a month to discuss the progress, problems, and solutions.

### Ethical consideration

We received approval from the institutional review board of Kathmandu University School of Medical Sciences, under Nepal Health Research Council (Reference number: 70/16) for the

human subject contacts and obtained written informed consent from the participants before collecting data for research purpose.

## Results

### Stakeholder identification

Based on the 7P framework, the identified stakeholders are presented in Fig 1. The identified stakeholders were: customers or employees (patient and the public; purchasers); cafeteria managers, chef and staff (providers); hospital administration and finance department (payers; policy makers); human resource department (policy maker); cafeteria employee, bakery and rice mills (product maker); and research team members (principal investigators).

### Outcome of stakeholder engagement

Table 1 presents the stakeholder engagement in different activities within different study phases.

### During formative study phase

**Formation of study team.** We formed a study team representing stakeholders from each category as investigator or an advisor. Investigators directly participated in the day-to-day implementation of the study while advisors provided suggestion and guidance when required [18]. Four employees consisting of a medical doctor, a laboratory personnel, a pharmacist, and a community programs expert jointed the team as co-investigators. They brought their scientific expertise and unique perspective as a customer of cafeteria. The hospital administrator, a cafeteria manager and a nutritionist joined as advisor.

**Focus group discussions.** Majority of participants considered that the foods that were available in the hospital cafeteria were healthy. However, items such as confectionaries like cream donut, oily curry and meat, soda drinks, instant noodles etc. were considered unhealthy food. They emphasized that the foods were hygienically prepared and cheaper compared to hotels outside. Participants highlighted that the changes to improve food quality will be well received positively because most of them are health professionals. The most reported factors that would facilitate healthy eating were: (a) addition of healthy food options, (b) replacement of unhealthy foods with healthy options, and (c) appealing presentation of healthy foods. Active involvement of canteen management and administrator in the process of change was highly emphasized. The participants suggested to advertise healthy food options and educate both cafeteria operators and consumers on healthy eating. The major barriers to healthy eating were: (a) unavailability of healthy options, (b) lack of human resources to prepare healthy food, and (c) high price of healthy food and (d) food preferences. They suggested to add the healthy items which will demand less time and resources like adding the automated machines such as roti makers. The support staff commented that they might not be consuming the fruits even if they are added because of the high cost. However, health professionals expressed their willingness to pay more for access to healthy food. The participants were concerned that it would be difficult to change food preferences as the consumers prefer spicy and oily foods. Some of the unhealthy foods are greatly loved such as instant noodles, samosas, cream donuts, soda drinks etc., and changing food habit is challenging.

**In-depth interviews.** Cafeteria operator commented that the higher-level authorities should be involved in making healthy changes, deciding the menu, and fixing the price. One of the canteen managers highlighted the need of a committee to involve the canteen operator, administrative staff, medical doctors, and nutritionist to decide on the change in food menu,

**Table 1. Stakeholder engagement in different activities within different study phases.**

| Study phases | Activities | Stakeholder involved |
|---|---|---|
| **During Formative phase** | Formation of study team | • Principal investigator (Research team members)<br>• Purchasers (Employees including a medical doctor, laboratory personnel, a pharmacist, and a community programs expert)<br>• Payers (The Hospital administrator)<br>• Policy maker (The Hospital administrator, A nutritionist)<br>• Provider (A cafeteria manager) |
| | Focus Group Discussions | • Principal investigator (Research team members)<br>• Purchasers (Employees) |
| | In-depth interviews | • Provider (Cafeteria operator, Cafeteria manager)<br>• Product maker (Cafeteria Staffs, Bakery and Rice mills)<br>• Policy maker (Human resource manager) |
| **During development of research design and intervention phase** | Workshops | • Principal investigator (Research team members)<br>• Provider (Cafeteria operator, Cafeteria manager)<br>• Product maker (Cafeteria Staffs)<br>• Purchasers (Employees) |
| | Individual consultation | • Principal investigator (Research team members)<br>• Purchasers (Employees including medical doctor, laboratory personnel, pharmacist, and community programs expert)<br>• Payers (Hospital administrator)<br>• Policy maker (Hospital administrator, Nutritionist)<br>• Provider (Cafeteria operator, Cafeteria manager) |
| **During testing of intervention phase** | Meeting of study team | • Principal investigator (Research team members)<br>• Purchasers (Employees including a medical doctor, laboratory personnel, a pharmacist, and a community programs expert)<br>• Payers (Hospital administrator)<br>• Policy maker (Hospital administrator, Nutritionist)<br>• Provider (Cafeteria operator, Cafeteria manager) |
| | Information sessions | • Principal investigator (Research team members)<br>• Policy makers (Departmental head and supervisors)<br>• Purchasers (Employees) |
| | Formation of cafeteria intervention committee: | • Product maker (Chef, Canteen operator)<br>• Policy makers (Administrative staffs, Nutritionist)<br>• Purchasers (Employees including medical doctors) |

price, and to monitor the availability of healthy options in canteen. The canteen operators mentioned that the hospital cafeteria provides food in subsidized price and does not intend to make profit. The operators were concerned about the lack of knowledge on healthy eating among the cafeteria staff and pointed that providing health education to them could facilitate making changes. They mentioned that the cafeteria staff receive training on hygiene occasionally but have never received training on healthy eating and healthy cooking. The major barrier for healthy eating reported by canteen operator were: (a) lack of adequate human resource to add food options; (b) lack of knowledge on the healthy cooking among cafeteria staff, (c) unavailability of healthy food options in the cafeteria and (d) no food supply for healthy foods such as brown rice, brown bread, organic vegetables etc. The canteen operator reported that they are not well-staffed to provide healthy foods such as whole wheat pan bread (roti), fruits and salad as their preparation is labor intensive. The human resource manager thought that increasing efficiency of the available staff was more important than adding staff. The

stakeholders contributed to the interpretation of qualitative data and contextualize the findings. The details of the FGD and Interview results are available elsewhere [19]. In Brief, four focus group discussions of 33 participants and nine in-depth interviews among six canteen operators and three managers identified the availability of affordable healthy food options in the cafeterias, a commitment to such promotion by the cafeteria manager, operators, staff and hospital administration and the level of education of the employees, as the factors to promote healthy eating. In addition, the unavailability of healthy options, including the lack of food supply from local market, the higher cost of healthy foods, individual food preferences and limited human resources in the cafeteria were identified as barriers for healthy eating [19].

## During development of research design and intervention phase

**Workshops.** The discussions in the workshop were very interactive and productive. The cafeteria managers and cafeteria staffs emphasized the need for active involvement of the departments of finance and administration because these departments had major role deciding the menu and price of the food served in the cafeteria. This led to formation of canteen intervention committee with chairmanship of the hospital administrator and representatives from finance, human resource, nutrition, employee, and researchers. There was univocal agreement on the need of adding healthy food items. However, lack of human resource in the cafeteria was a major challenge. In response, automated roti (whole wheat pan bread) maker was suggested. The details on which food to add and which to remove were discussed and finalized in the workshop. In addition, the pre-post design was suggested in contrast to randomized trial to test the effectiveness of the changes in cafeteria on cardiometabolic risk because it would be unethical to provide healthy food items in some cafeterias only. Additionally, the risk of contamination was emphasized as customers do not exclusively go to one cafeteria. Consumers and chefs raised the concern on acceptability of brown rice in place of white rice. Hence, a brown rice tasting experiment was suggested.

**Individual consultation.** Based on the findings of the FGDs, in-depth interviews and workshops, the study team prepared the description of the cafeteria intervention and proposed study design to test the effectiveness of the intervention on cardiometabolic risks. The proposal was described and discussed in-person with the stakeholders. Lack of comparison group was indicated as a weakness of the study in the initial pre-post design. As a result, six months of control period was added before intervention was implemented to be able to compare outcome measures with and without the intervention.

## During testing of intervention phase

**Meeting of study team.** In the weekly meeting of the study team, the implementation strategies were discussed and finalized in detail. This included details on disseminating information on study, call for participation, informed consent administration, data collection from the participants, enrolment planning, and intervention start dates. The internal insights of employees in the study team members guided the decision making. When the effectiveness study started, the progress on enrollment, problems and strategies were discussed weekly. For example, when there was slow blood sample collection, the representative from the laboratory suggested to provide breakfast voucher to the participants. Adding breakfast voucher worth 70 cents in subsequent blood draw appointment speeded up the sample collection. Similarly, it was difficult to enroll the nurses who had night duties. The doctor in the study team suggested to consider nurse's duty roster and contact nurses immediately after completion of duty hours in the morning. This strategy helped to increase the participation of the nurses in the study.

**Information sessions.**   The information session helped to reach and interact with wider employee group. The department head provided date and venue for the information session. During the information session, feedback was obtained on the design of the study and participant recruitment. A major issue that came up was the involvement of supervisors in participants' recruitment. The employees commented that presence of supervisors during the information session might be perceived as encouragement from them to participate. To make the participation voluntary, we conducted the sessions in absence of higher-level staff in each department, the identifiers of the participants were collected separately and were not disclosed to anyone.

**Formation of cafeteria intervention committee.**   The cafeteria intervention committee had key decisive role in designing, implementing, monitoring, and evaluating the cafeteria intervention. Each component of the intervention was discussed and decided by the intervention committee after weighing their benefits, financial implications, and perceived acceptance by the employee. The committee developed a monitoring sheet that the nutritionist and a research team member filled every week to measure the adherence to the defined intervention in all four cafeterias. The monitoring team also interacted with the chefs, cafeteria staffs and customers informally and collected their concerns. The monitoring data were assessed, and concerns were presented to the committee every month. The committee discussed the problems and strategies to address them. For example, in the initial week, there was problem of supply of whole grains such as brown rice and brown bread. The committee approached the local mill and a local bakery and ordered brown rice and brown bread for the cafeteria. After adding fruits on the menu, the sales of fruits in the first two months were low. The committee decided to launch a week-long campaign to promote fruits in each cafeteria which boosted their sales.

## Benefits and challenges of stakeholder engagement

We are confident that our project, the project goals, methods, and intervention have evolved positively based on our collaborations with stakeholders. Most importantly, their input was necessary in determining and describing cafeteria intervention. Their continuous involvement in monitoring helped in solving problems promptly and enhanced adherence to intervention. In addition, their input shaped methods, plans and processes for recruiting participants and data collection.

While we had a positive experience of involving stakeholder, it was not devoid of challenges. A major challenge was time availability of stakeholders as they were busy in their respective works. To address this, we scheduled meetings according to the availability of the stakeholders. We had to conduct multiple workshops and information sessions to accommodate the time of cafeteria staff and hospital staff. This was time and resource consuming. It was also important to allow time for stakeholders to get acquainted with their new roles associated with the intervention. Their knowledge of a healthy diet was dominated by the cultural norms and there was some misconception. So, we invested time and resources in training them on healthy eating. This also led to initiation of positive changes in cafeterias during the control period before the predetermined intervention date. We also felt that research priorities do not necessarily meet the priorities of the stakeholders. For example, the cafeteria staff emphasized more on the changes in physical structures such as adding culinary. There was language barrier, as researchers were comfortable using scientific jargons such as 'epidemiology' 'control' 'confounding', 'subjects' etc. making the stakeholders confused at times. It was difficult to translate or interpret these works well in local language. Additionally, the food beliefs of researchers and stakeholders were different. Stakeholders strongly believe in eating warm

'white rice' twice a day. In addition, fruits were not considered appropriate to take during lunch. Knowledge sessions on healthy diets addressed these deep rooted beliefs to some extent but changing food belief system takes longer time and more effort [6, 20].

## Discussion

We identified employees, cafeteria managers, workplace administrators, finance manager, human resource manager, cafeteria staff and researchers as major stakeholders in the Nepal pioneer worksite intervention study to prevent cardiometabolic risks. The stakeholders were included in the study team, and they brought their expertise in the subject matter as well as insights as employees and customers of the cafeteria. The exploration of the facilitators and barriers to healthy eating at their worksite provided a base for defining the intervention and study attributes. Interactive workshops provided opportunity and mandate for the stakeholders to participate in decision making and strengthened the investigator-stakeholder relationship. Individual consultations helped get stronger buy-in and commitment from the higher authorities and employees. Regular meeting of the study team and canteen intervention committee guaranteed continuous engagement in problem solving and decision making. It reinforced ownership and enhanced project sustainability. The major challenges were the time-consuming nature of the process, language barriers, and differences in health and food beliefs between researchers and stakeholders.

We mapped and engaged 7 different types of stakeholders in our study. Patient-centered outcome research projects have reported engaging clinicians, caregivers, advocacy organizations and health system representatives; but only few engaged payers and policy makers [13, 21]. We incorporated the views of multiple types of stakeholders into our study during formative phase, development of intervention and study and execution of the study. For example, we formed cafeteria intervention committee based on the suggestion of cafeteria manager in in-depth interview. Brown-rice tasting experiment was also conducted to solve the concern of the participants of workshops (consumers and chefs) on the acceptability of brown rice. Engagement has most frequently been reported in developing research questions [9]. This study described the effort, attention, resources and flexibility for concerted engagement emphasizing on meaningful and continuous collaboration, shared leadership and decision-making power, and adaptation to the needs of stakeholders. Other researchers have highlighted the importance of sharing power for successful research engagement [22].

The input from different stakeholder influenced our study goals, intervention, methods, and materials. This enhanced the appropriateness and relevance in the local context. Similar experience was reported by other investigators [6]. Making research more beneficiaries-centric, culturally relevant and accessible to potential research participants have been reported to refine proposals, recruitment materials, and participant compensation [23]. In our project, the stakeholders contributed to the interpretation of qualitative data and the findings, similar to the study by Boote J et al [11]. This is consistent with the available evidence on positive impacts of engaged research [6, 12].

The stakeholder engagement has contributed positively in our study from formative phase to execution phase. Various levels of stakeholder involvement, such as consultation, collaboration or user-control has been proposed [24]. However, caution has been recommended against assuming that higher levels of involvement is always better [25]. Some researchers in a national survey from UK claimed that decisions about the appropriateness of consumer involvement in research should be made on a case-by-case basis [13].

The major challenges that we faced were the time-consuming nature of the process, language barriers, and differences in health and food beliefs between researchers and stakeholders.

Similar challenges have been reported in other studies [6, 20]. Lack of time—both for stakeholders and researchers—was the most frequently noted challenge and was identified as a top barrier to engagement in a recent survey of investigators [20]. Thus, it is crucial to develop strategies that maximize stakeholder input with minimum time and resources [26].

The generalizability of our findings is limited because of the unique setting where the research was conducted. Because it was a university hospital, we had an advantage of including the employees with specific research expertise as co-investigators. The stakeholders had more interest in and experience with the health interventions and research. Thus, their knowledge and attitude about engagement may be more positive.

## Conclusion

It was possible to engage and benefit from stakeholder's engagement on the design, implementation and monitoring of a workplace-based hypertension and diabetes management research program in Nepal. We recommend ongoing program development and implementation by stakeholder input. In future, we plan to evaluate the influence of stakeholders on the research project and assess the engagement and effectiveness through their perspective, as well as longer term outcomes on health decision making beyond the conduction of research.

## Supporting information

**S1 Dataset.**
(XLSX)

## Acknowledgments

We would like to acknowledge the Dhulikhel Hospital—Kathmandu University Hospital for their permission and cooperation to conduct this study. We appreciate the contribution of all the study participants.

## Author Contributions

**Conceptualization:** Archana Shrestha, Dipesh Tamrakar, Donna Spiegelman.

**Data curation:** Archana Shrestha, Bhawana Shrestha.

**Formal analysis:** Archana Shrestha, Bhawana Shrestha, Biraj Man Karmacharya, Abha Shrestha, Prajjwal Pyakurel.

**Funding acquisition:** Archana Shrestha, Dipesh Tamrakar, Donna Spiegelman.

**Project administration:** Archana Shrestha, Dipesh Tamrakar.

**Supervision:** Donna Spiegelman.

**Validation:** Donna Spiegelman.

**Writing – original draft:** Archana Shrestha.

**Writing – review & editing:** Archana Shrestha, Dipesh Tamrakar, Bhawana Shrestha, Biraj Man Karmacharya, Abha Shrestha, Prajjwal Pyakurel, Donna Spiegelman.

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
