## [Decision Letter · Decision Letter 0]

27 Apr 2021

PONE-D-21-05456

Stakeholder engagement in a hypertension and diabetes prevention research program – Description and Lessons learned

PLOS ONE

Dear Dr. Shrestha,

Thank you for submitting your manuscript to PLOS ONE. After careful consideration, we feel that it has merit but does not fully meet PLOS ONE’s publication criteria as it currently stands. Therefore, we invite you to submit a revised version of the manuscript that addresses the points raised during the review process.

We look forward to receiving your revised manuscript.

Kind regards,

Chaisiri Angkurawaranon

Academic Editor

PLOS ONE

Journal Requirements:

2. When reporting the results of qualitative research, we suggest consulting the COREQ guidelines: http://intqhc.oxfordjournals.org/content/19/6/349

In this case, please consider including more information on the number of interviewers, their training and characteristics; how participants were selected; if a pilot study was tested; how data was coded; if bias issues were considered.

Moreover, please provide the interview guide used as a Supplementary file.

3. Please ensure you have included the registration number for the clinical trial referenced in the manuscript.

Reviewers' comments:

Reviewer's Responses to Questions

**Comments to the Author**

1. Is the manuscript technically sound, and do the data support the conclusions?

Reviewer #1: Yes

Reviewer #2: Yes

2. Has the statistical analysis been performed appropriately and rigorously? 

Reviewer #1: N/A

Reviewer #2: N/A

3. Have the authors made all data underlying the findings in their manuscript fully available?

Reviewer #1: No

Reviewer #2: No

4. Is the manuscript presented in an intelligible fashion and written in standard English?

Reviewer #1: Yes

Reviewer #2: Yes

5. Review Comments to the Author

Reviewer #1: This manuscript describes stakeholder engagement activities and lessons learned in a diabetes and hypertension prevention study in workplace settings in Nepal. The authors describe stakeholder engagement activities during the formative, intervention development, and study conduct phases of this project. In general, this paper could benefit from grammatical review as there are several errors throughout. Additionally, the literature review should be updated to reflect current gaps, as there are several existing articles to date that offer engagement implementation examples, which was stated as a gap and strength of current paper. The impact of stakeholder engagement on research activities and translation is an area for further contribution and an opportunity where this manuscript could offer insight. Areas of stakeholder impact should be more clearly identified in the paper. In addition to above, consider the following to strengthen this manuscript:

-Consider re-designing Table 1 into a Figure and move to first presentation of stakeholders using 7P Framework in Methods section (not in Results section as current placement).

-Although it was stated that results from qualitative work is under consideration for publication elsewhere, this section could benefit from including more specifics regarding questions asked during focus group sessions/in-depth interviews with stakeholders and/or qualitative supporting quotes from discussions, since these informed intervention. Who came up with interview questions? How did the research team decide which feedback to implement?

-Consider creating a new Table 1 in Results that includes stakeholder type and the actual activities that the stakeholders were engaged in (i.e. formative and then subheading for which activity i.e. focus group). It was not clear within text the myriad of stakeholders who participated in the various study processes. This table could also include a column "Impact" to highlight how stakeholder contributions informed the study and in what way(s).

Reviewer #2: Thank you for the opportunity to review this manuscript. Herewith my comments

Overall

An important work in the area stakeholder engagement. Generally, well written but needs some language editing and improvement in flow in some sections.

Abstract

Lines 39-41. What is described does not include interpreting the results and results dissemination.

Lines 47: I am not sure increased trust in research and confidence with the researchers is a result or what can be inferred (or assumed) given the processes and outcomes. What evidence do you have for this allusion?

Line 54: The manuscript has not reported on engaging the stakeholders in the evaluation may be monitoring.

Introduction

Line 62: Give reference

See comment on line 67

Materials and methods

-Line 88: Aim would be more appropriate than “theme”

-Sub-titles e.g., Phase 1 Formative study would be better be presented as Phase 1: Formative study

-Lines 106 to lines 202: Flow in stakeholder engagement needs to be improved for easier reading. I suggest a main sub-title “Stakeholder engagement” and smaller sub-titles as follows: stakeholder definition, engagement objectives, stakeholder identification and stakeholder methods of engagement.

-Not sure it is necessary to number the phases e.g., line 134

-Lines 135-137. What criteria did you use to identify/include the individuals from each stakeholder category after conducting the mapping?. This also applies to those included in focus groups discussion (FDGs).

-Purposive selection for interviews- what specific information did you expect that these participants could be having unlike others

-I think its important to mention qualitative methods were used for FDGs and the in depth interviews- as the methods holds specific benefits and give support from literature on the choice of method

-Line 172: I think topic rather than theme will be more appropriate

-What was the stakeholders’ role in the workshops?

Results

-Line 211: What informed/guided the inclusion to a certain category (investigator vs. advisor). Any evidence from literature?

-As a reader, I was expecting more details of the FDGs and interviews, but I saw at the end the statement that the results are under consideration for publication elsewhere. Just wondering, the implication of this given that data needs to be availed by authors.

-Lines 231-232: They suggested to add the healthy items which will demand less time and resources like adding the automated machines such as roti makers. Rephrase I don’t think automated machines needs less resource (may be human but not financial)

-Interchange order of sentences on lines 218-219 and lines 219-221 for better flow.

-What was the outcome of the food testing and how did it inform the intervention?

-Rephrase lines 356-358 for clarity

-Lines 392-394: There is nowhere that is reported how the stakeholders were involved in interpreting the data.

Discussion

Lines 379 to 387: It would be useful if the authors could highlight on how they incorporated the views of the stakeholders in each step, probably in a Table format. For example; What was the final goal and what stakeholder input lead to that outcome.

I think it is important for the reader to know where the project is at the moment- is still being implemented and prospect of sustainability given that its one of the objective of involving stakeholders.

*There is some restriction to the data. Wondering about the qualitative data that is to be published elsewhere and part is included in the current manuscript.

Some comments are in the manuscript itself

6. PLOS authors have the option to publish the peer review history of their article (what does this mean?). If published, this will include your full peer review and any attached files.

Reviewer #1: No

Reviewer #2: **Yes: **Jane W Muchiri

---

## [Author Response · Author response to Decision Letter 0]

8 Jun 2022

Subject: Re-submission of an original research article to PLOS ONE journal addressing the reviewer's comment

Dear sir/madam,

It is our pleasure to re-submit to you and the editorial teams our manuscript entitled “Stakeholder Engagement in a Hypertension and Diabetes Prevention Research Program – Description and Lessons Learned”. The paper aims to describe the stakeholder engagement process and the lessons learned from this process while designing, implementing and monitoring a study on diabetes and hypertension prevention in workplace settings in Nepal. It provides important insights for future researchers that aim to engage stakeholders in national-level preventive programs in health.

We would like to thank all the reviewers for their constructive comments. The responses to the reviewer’s comments are addressed individually below:

Reviewer 1

This manuscript describes stakeholder engagement activities and lessons learned in diabetes and hypertension prevention study in workplace settings in Nepal. The authors describe stakeholder engagement activities during the formative, intervention development, and study conduct phases of this project. In general, this paper could benefit from a grammatical review as there are several errors throughout. Additionally, the literature review should be updated to reflect current gaps, as there are several existing articles to date that offer engagement implementation examples, which was stated as a gap and strength of current paper. The impact of stakeholder engagement on research activities and translation is an area for further contribution and an opportunity where this manuscript could offer insight. Areas of stakeholder impact should be more clearly identified in the paper. In addition to above, consider the following to strengthen this manuscript:

-Consider re-designing Table 1 into a Figure and move to first presentation of stakeholders using 7P Framework in Methods section (not in Results section as current placement).

>> The Table 1 is converted to figure and shifted to methods section

-Although it was stated that results from qualitative work is under consideration for publication elsewhere, this section could benefit from including more specifics regarding questions asked during focus group sessions/in-depth interviews with stakeholders and/or qualitative supporting quotes from discussions, since these informed intervention. Who came up with interview questions? How did the research team decide which feedback to implement?

>> The result and methods are available in detailed in the paper below and it is now cited.

Tamrakar D, Shrestha A, Rai A, Karmacharya BM, Malik V, Mattei J, et al. Drivers of healthy eating in a workplace in Nepal: a qualitative study. BMJ Open. 2020;10: e031404. doi:10.1136/bmjopen-2019-031404

-Consider creating a new Table 1 in Results that includes stakeholder type and the actual activities that the stakeholders were engaged in (i.e. formative and then subheading for which activity i.e. focus group). It was not clear within text the myriad of stakeholders who participated in the various study processes. This table could also include a column "Impact" to highlight how stakeholder contributions informed the study and in what way(s).

>>Table is generated as per your suggestions

Reviewer 2

An important work in the area stakeholder engagement. Generally, well written but needs some language editing and improvement in flow in some sections.

Abstract

Lines 39-41. What is described does not include interpreting the results and results dissemination.

>> It is revised as per your suggestion

Lines 47: I am not sure increased trust in research and confidence with the researchers is a result or what can be inferred (or assumed) given the processes and outcomes. What evidence do you have for this allusion?

>> It is revised as per your suggestion

Line 54: The manuscript has not reported on engaging the stakeholders in the evaluation may be monitoring.

>>The comment is addressed 

Introduction

Line 62: Give reference

>> Reference is added

See comment on line 67

>> comment addressed

Materials and methods

-Line 88: Aim would be more appropriate than “theme”

>> “theme” is changed to “aim”

-Sub-titles e.g., Phase 1 Formative study would be better be presented as Phase 1: Formative study

-Lines 106 to lines 202: Flow in stakeholder engagement needs to be improved for easier reading. I suggest a main sub-title “Stakeholder engagement” and smaller sub-titles as follows: stakeholder definition, engagement objectives, stakeholder identification, and stakeholder methods of engagement.

>> Its flow is changed according to the suggestion provided.

-Not sure it is necessary to number the phases e.g., line 134

>>The phrases are now un-numbered

-Lines 135-137. What criteria did you use to identify/include the individuals from each stakeholder category after conducting the mapping?. This also applies to those included in focus groups discussion (FDGs).

>> Added to the manuscript. We selected adults aged 18 years or older, employee of the hospital, who provided informed consent in each stakeholder category. 

-Purposive selection for interviews- what specific information did you expect that these participants could be having unlike others

>> They had unique experience of making first hand decisions and implementation at the cafeteria and other wellness programs for the employees. These interviewees brought their perspectives on the decision making system, how it affected healthy food choices in the cafeteria and other administrative decisions regarding employee’s health. This would provide insights into the facilitators and barriers to healthy eating in the workplace environment.

-I think its important to mention qualitative methods were used for FDGs and the in-depth interviews- as the methods hold specific benefits and give support from the literature on the choice of method

>> >> The result and methods for FGDs and in depth interviews are available in detailed in the paper below and it is now cited.

Tamrakar D, Shrestha A, Rai A, Karmacharya BM, Malik V, Mattei J, et al. Drivers of healthy eating in a workplace in Nepal: a qualitative study. BMJ Open. 2020;10: e031404. doi:10.1136/bmjopen-2019-031404

-Line 172: I think topic rather than theme will be more appropriate

>> The word theme is replaced by the topic

-What was the stakeholders’ role in the workshops?

>>The stakeholders provided their inputs in decision making. We presented the initial findings with the stakeholders and collected their feedback in the interpretation of the results.

Results

-Line 211: What informed/guided the inclusion to a certain category (investigator vs. advisor). Any evidence from literature?

>> Investigator were those who directly participated in the day-to-day implementation of the study, compared to advisors who provided their suggestions and guidances when asked for.

-As a reader, I was expecting more details of the FDGs and interviews, but I saw at the end the statement that the results are under consideration for publication elsewhere. Just wondering, the implication of this given that data needs to be availed by authors.

>>>> The result and methods are available in detailed in the paper below and it is now cited.

Tamrakar D, Shrestha A, Rai A, Karmacharya BM, Malik V, Mattei J, et al. Drivers of healthy eating in a workplace in Nepal: a qualitative study. BMJ Open. 2020;10: e031404. doi:10.1136/bmjopen-2019-031404

-Lines 231-232: They suggested to add the healthy items which will demand less time and resources like adding the automated machines such as roti makers. Rephrase I don’t think automated machines needs less resource (may be human but not financial)

>> The sentence is rephrased as “They suggested to add the healthy items which will demand less time and human resources like adding the automated machines such as roti makers.”

-Interchange order of sentences on lines 218-219 and lines 219-221 for better flow.

>> The sentences are interchanged

-What was the outcome of the food testing and how did it inform the intervention?

>> Added to the manuscript. Participant’s perspectives on brown rice was found to be changed after tasting. Participants were willing to switch to brown rice as they found that brown rice had higher quality.[17]

-Rephrase lines 356-358 for clarity

>>The line is rephrased for clarity

-Lines 392-394: There is nowhere that is reported how the stakeholders were involved in interpreting the data.

>> The comment is addressed in the manuscript.

Discussion

Lines 379 to 387: It would be useful if the authors could highlight on how they incorporated the views of the stakeholders in each step, probably in a Table format. For example; What was the final goal and what stakeholder input lead to that outcome.

>> We added some ways that we used to incorporate views of stakeholders. (. For example, we formed cafeteria intervention committee based on the suggestion of cafeteria manager in in-depth interview. Brown-rice tasting experiment was also conducted to solve the concern of consumers and chefs on the acceptability of brown rice in workshop.)

Other comments 

1. Please ensure you have included the registration number for the clinical trial referenced in the manuscript.

The registration number is added in the manuscript. This study was registered on clincaltrials.gov (Identification Member:NCT03447340; Registration date: 27th February, 2018).

2. To help ensure that the wording of your manuscript is suitable for publication, would you please also add this statement at the beginning of the Methods section of your manuscript file.

The ethical statement is added in the publication. 

Ethical consideration: We received approval from the institutional review board of Kathmandu University School of Medical Sciences and Nepal Health Research Council for the human subject contacts and obtained written informed consent from the participants before collecting data for research purpose.

3. Data availability statement. 

The dataset is uploaded is OSF. The link to dataset is https://osf.io/bqsrd/

I think it is important for the reader to know where the project is at the moment- is still being implemented and prospect of sustainability given that its one of the objective of involving stakeholders

We look forward to your comments and remain with best regards;

---

## [Decision Letter · Decision Letter 1]

15 Jul 2022

PONE-D-21-05456R1Stakeholder engagement in a hypertension and diabetes prevention research program – Description and Lessons learnedPLOS ONE

Dear Dr. Shrestha,

Thank you for submitting your manuscript to PLOS ONE. After careful consideration, we feel that it has merit but does not fully meet PLOS ONE’s publication criteria as it currently stands. Therefore, we invite you to submit a revised version of the manuscript that addresses the points raised during the review process. Please submit your revised manuscript by Aug 29 2022 11:59PM. If you will need more time than this to complete your revisions, please reply to this message or contact the journal office at plosone@plos.org. Please include the following items when submitting your revised manuscript:A rebuttal letter that responds to each point raised by the academic editor and reviewer(s). You should upload this letter as a separate file labeled 'Response to Reviewers'.A marked-up copy of your manuscript that highlights changes made to the original version. You should upload this as a separate file labeled 'Revised Manuscript with Track Changes'.An unmarked version of your revised paper without tracked changes. You should upload this as a separate file labeled 'Manuscript'.If applicable, we recommend that you deposit your laboratory protocols in protocols.io to enhance the reproducibility of your results. Protocols.io assigns your protocol its own identifier (DOI) so that it can be cited independently in the future. For instructions see: https://journals.plos.org/plosone/s/submission-guidelines#loc-laboratory-protocols. Additionally, PLOS ONE offers an option for publishing peer-reviewed Lab Protocol articles, which describe protocols hosted on protocols.io. Read more information on sharing protocols at https://plos.org/protocols?utm_medium=editorial-email&utm_source=authorletters&utm_campaign=protocols.

We look forward to receiving your revised manuscript.

Kind regards,

Chaisiri Angkurawaranon

Academic Editor

PLOS ONE

Journal Requirements:

Reviewers' comments:

Reviewer's Responses to Questions

**Comments to the Author**

1. If the authors have adequately addressed your comments raised in a previous round of review and you feel that this manuscript is now acceptable for publication, you may indicate that here to bypass the “Comments to the Author” section, enter your conflict of interest statement in the “Confidential to Editor” section, and submit your "Accept" recommendation.

Reviewer #2: (No Response)

2. Is the manuscript technically sound, and do the data support the conclusions?

Reviewer #2: Partly

3. Has the statistical analysis been performed appropriately and rigorously? 

Reviewer #2: N/A

4. Have the authors made all data underlying the findings in their manuscript fully available?

Reviewer #2: Yes

5. Is the manuscript presented in an intelligible fashion and written in standard English?

Reviewer #2: Yes

6. Review Comments to the Author

Reviewer #2: 1. General

An important topic that will contribute to stakeholder engagement in interventions knowledge base. But unless, I have not understood the title “It is description & lessons learnt” As a reader I don’t think I still can clearly pinpoint the lessons learnt unless in the manuscript it is equivalent to benefits & challenges. Wonder if the title should not be Description & experiences?

Generally well written. However, the manuscript can benefit from some editing e.g. Line 143 word interview should not be started with upper case. Line 144: Add “a” between guide and in private room. More information on specific places to correct are given in comments on the manuscript itself.

I see some of the work mentioned in the manuscript has been published, for example the FDGs & in-depth interviews. I think it is important that the reader understands from the word go, that details are reported somewhere. For example, I had a question on the number of participants per FDG, and wondering about the participants quotes, but I see these details are in reference 19.

I think what would important is to highlight what informed the authors decisions- for example to form a study committee, use focus groups/workshops and not any methods for engagement, in case a reader would want to conduct a similar stakeholder engagement but still briefly report how these activities were done. I believe you were guided by literature.

2 Introduction: The statement “New scientific evidence” appears incomplete without saying what kind of evidence. For example, is it health promotion/medical intervention effectiveness?

3 Methods

Lines 122- 126: The formation of the study team is unclear. I would believe some stakeholder categories had more than one person. How did you decide which representative(s) to approach? Are there people who declined to take part, since you have said those who provided informed consent? The last part “The stakeholders represented as co-investigators and advisors” need to be rephrased for clarity. [Probably referring to your published work could help as well].

I think its important to mention the rationale for the constituents of study team in the methods section.

Workshops: As a reader I will be interested to know the aim of the workshop.

It may be good to give the ethics approval number

The authors should detail how stakeholders were involved in the interpretation of the results.

Line 352- put a reference as this is documented in the literature.

I believe the first part of the discussion i.e 356 to 367, there would be literature documenting these strategies in other studies. If not the authors should indicate there is no other studies identified that have used the same strategies as the current study.

Conclusion does not include any comment on lessons learnt.

7. PLOS authors have the option to publish the peer review history of their article (what does this mean?). If published, this will include your full peer review and any attached files.

Reviewer #2: No

---

## [Author Response · Author response to Decision Letter 1]

22 Aug 2022

Response to Reviewers

08/22/2022

To

Editors and team

PLOS ONE journal

Subject: Re-submission of an original research article to PLOS ONE journal addressing the reviewer's comment

Dear sir/madam,

It is our pleasure to re-submit our manuscript entitled “Stakeholder Engagement in a Hypertension and Diabetes Prevention Research Program” to the editorial teams. The paper aims to describe the stakeholder engagement process and the lessons learned from this process while designing, implementing and monitoring a study on diabetes and hypertension prevention in workplace settings in Nepal. It provides important insights for future researchers that aim to engage stakeholders in national-level preventive programs in health.

We would like to thank all the reviewers for their constructive comments. The responses to the reviewer’s comments are addressed individually below:

 Reviewer 2

General:

An important topic that will contribute to stakeholder engagement in interventions knowledge base. But unless, I have not understood the title “It is description & lessons learnt” As a reader I don’t think I still can clearly pinpoint the lessons learnt unless in the manuscript it is equivalent to benefits & challenges. Wonder if the title should not be Description & experiences?

>> Based on your suggestions we removed Description & experiences from title.

Generally, well written. However, the manuscript can benefit from some editing e.g. Line 143: word interview should not be started with upper case. Line 144: Add “a” between guide and in private room. More information on specific places to correct are given in comments on the manuscript itself.

>>Edited based on your comments

I see some of the work mentioned in the manuscript has been published, for example the FDGs & in-depth interviews. I think it is important that the reader understands from the word go, that details are reported somewhere. For example, I had a question on the number of participants per FDG, and wondering about the participant’s quotes, but I see these details are in reference 19.

>> We have added brief of the methods and results 

I think what would important is to highlight what informed the authors decisions- for example to form a study committee, use focus groups/workshops and not any methods for engagement, in case a reader would want to conduct a similar stakeholder engagement but still briefly report how these activities were done. I believe you were guided by literature.

>> It is mentioned in under Engagement objectives heading.

2 Introduction: The statement “New scientific evidence” appears incomplete without saying what kind of evidence. For example, is it health promotion/medical intervention effectiveness?

>> Addressed

Methods:

Lines 122- 126: The formation of the study team is unclear. I would believe some stakeholder categories had more than one person. How did you decide which representative(s) to approach? Are there people who declined to take part, since you have said those who provided informed consent? The last part “The stakeholders represented as co-investigators and advisors” need to be rephrased for clarity. [Probably referring to your published work could help as well]. I think its important to mention the rationale for the constituents of study team in the methods section

>> The formation of the study team is now rephrased and edited to make more clear statement

Workshops: As a reader I will be interested to know the aim of the workshop. It may be good to give the ethics approval number The authors should detail how stakeholders were involved in the interpretation of the results.

>>Aim of workshop and approval number are added in the manuscript

Line 352- put a reference as this is documented in the literature. I believe the first part of the discussion i.e. 356 to 367, there would be literature documenting these strategies in other studies. If not the authors should indicate there is no other studies identified that have used the same strategies as the current study.

>>Reference is added to the line 352.

Conclusion does not include any comment on lessons learnt.

>> We removed lesson learnt from title. And hence lesson learnt is not placed in the conclusion section.

We look forward to your comments and remain with best regards;

Yours sincerely,

 Dr. Archana Shrestha

Department of Public Health

Kathmandu University School of Medical Sciences

Dhulikhel, Nepal

---

## [Decision Letter · Decision Letter 2]

15 Sep 2022

PONE-D-21-05456R2Stakeholder engagement in a hypertension and diabetes prevention research programPLOS ONE

Dear Dr. Shrestha,

Thank you for submitting your manuscript to PLOS ONE. After careful consideration, we feel that it has merit but does not fully meet PLOS ONE’s publication criteria as it currently stands. Therefore, we invite you to submit a revised version of the manuscript that addresses the points raised during the review process. The reviewer had some minor revisions request.  If you can attend to theses, I will make a decision without the need for further re-review.

We look forward to receiving your revised manuscript.

Kind regards,

Chaisiri Angkurawaranon

Academic Editor

PLOS ONE

Journal Requirements:

Additional Editor Comments:

The reviewer had some minor revisions request (attached file).  If you can attend to theses, I will make a decision without the need for further re-review.

Reviewers' comments:

Reviewer's Responses to Questions

**Comments to the Author**

1. If the authors have adequately addressed your comments raised in a previous round of review and you feel that this manuscript is now acceptable for publication, you may indicate that here to bypass the “Comments to the Author” section, enter your conflict of interest statement in the “Confidential to Editor” section, and submit your "Accept" recommendation.

Reviewer #2: All comments have been addressed

2. Is the manuscript technically sound, and do the data support the conclusions?

Reviewer #2: Yes

3. Has the statistical analysis been performed appropriately and rigorously? 

Reviewer #2: N/A

4. Have the authors made all data underlying the findings in their manuscript fully available?

Reviewer #2: Yes

5. Is the manuscript presented in an intelligible fashion and written in standard English?

Reviewer #2: Yes

6. Review Comments to the Author

Reviewer #2: Thank you for addressing the concerns. However, there are still some sentences which are still not clear as indicated in my comments in the draft manuscript attached which are important to address to improve your manuscript. My thoughts regarding the title: though appropriate to address the concern raised previously, it may no be as catchy to the audience as the previous one. All the best with your manuscript and further work in this field!

7. PLOS authors have the option to publish the peer review history of their article (what does this mean?). If published, this will include your full peer review and any attached files.

Reviewer #2: No

---

## [Author Response · Author response to Decision Letter 2]

29 Sep 2022

We would like to thank all the reviewers for their constructive comments. All comments are addressed and the track-changed files along with the clean manuscript files are submitted to the journal.

---

## [Editor Report · Decision Letter 3]

10 Oct 2022

Stakeholder engagement in a hypertension and diabetes prevention research program: description and lessons learned

PONE-D-21-05456R3

Dear Dr. Shrestha,

We’re pleased to inform you that your manuscript has been judged scientifically suitable for publication and will be formally accepted for publication once it meets all outstanding technical requirements.

Kind regards,

Chaisiri Angkurawaranon

Academic Editor

PLOS ONE
---

## [Editor Report · Acceptance letter]

12 Oct 2022

PONE-D-21-05456R3 

Stakeholder engagement in a hypertension and diabetes prevention research program: description and lessons learned 

Dear Dr. Shrestha:

I'm pleased to inform you that your manuscript has been deemed suitable for publication in PLOS ONE. Congratulations! Your manuscript is now with our production department. 

Kind regards, 

on behalf of

Dr. Chaisiri Angkurawaranon 

Academic Editor

PLOS ONE